# The Role of the Excluded

## Gianfranco Minati

Italian Systems Society, 20161 Milan, Italy; gianfranco.minati@airs.it

**Abstract:** We consider the peculiarity of unique events, such as those of a natural, evolutionary, and social nature. In particular, we consider unique social events that have had either the claim or the vocation of being salvific for humanity, such as the introduction over time of the Torah, Hinduism, Buddhism, Christianity, and Islam. We question how the claimed, general salvific vocation contrasts, or is inconsistent with, the non-retroactive temporality and locality of such events, which could not have happened otherwise. This undeclared and philosophically unsolved inconsistency then reappears in subsequent cultural contradictions and inadequacies, political and social allowances such as, for instance, homo-centrism and a pathological relation with Nature. In the case of Christianity, this inconsistency is represented by the painting reproduced in the article, a work in which the excluded humans and other living beings are represented as astonished by the occurrence in this moment, and in such an unnatural context. Furthermore, we consider the original understanding as related to concepts of classical physics, or of such concepts naively adopted within the texts considered sacred. However, in some religions, such as Christianity, the inconsistency is theologically solved. We stress the need to update the ancient original elementary, naïve, pre-classic philosophical and conceptual frameworks used so that these alleged inconsistencies and contradictions may be not only theologically solved, but also conceptually solved in more complex understandings of the world, for example, considering relativistic time, long-range interdependence, quantum entanglement, and theories of the universe. Without this update, the unique saving events can affect only religiously, that is, optionally, on the scientific and philosophical conceptions used. Without this adjustment, homo-centrist illusion and egoism prevail as the natural, linear consequential attitude without raising these questions. It rather assumes that the intervention is for involved human beings, and moreover for those who have had and are lucky enough to receive and practice it, ignoring the enormous inconsistency within the message itself, and its presumed general and available salvific nature. This requires theological, philosophical, and scientific interdisciplinarity. The theme concerns inconsistencies within and superficiality of the narratives and their treatment of the unique, salvific events, without any reference to possible general and retroactive effects of how these events are represented in the painting. We conclude that the subject should be debated by taking into account contemporary understandings, such as relativistic space and time, quantum physics, and of the universe, with new philosophical and anthropological approaches. This should be a matter of responsible philosophical and theological interdisciplinary debate involving science, suitable to establish new understandings.

**Keywords:** constructivism; homo-centrism; inconsistency; multiple roles; multiple systems; objectivism; reverse causality; unique events

## 1. Introduction

It is possible to consider events termed *unique* as having overall, long-range spatial and temporal effects when, for instance, those events are characterized by no or by very difficult and unlikely repeatability. In reality, considering adequate levels of description and scaling, *non-unique* events are of irrelevant number. However, the uniqueness of different events is often ignored, since they appear related to countless cases that have equivalent effects. The real uniqueness, to which we refer often in this article, relates to non-equivalent cases

having non-equivalent effects, and also to the enormity of characterizing both instantaneous and longer-term aftermath: the repercussions.

An example of natural, unique events is given by the occurring of ice ages of the Earth. For example, the first went from approximately 800 to 600 million years ago, in the so-called late Proterozoic period. It is assumed that the Earth was completely covered with ice. These are events with dates of occurrence and termination (duration), having levels of approximation, and with overall spatial effects, as usual in cosmology.

A daily example in Nature is the uniqueness of atmospheric events capable of unleashing microscopically unique thunderstorms, however equivalent their macroscopic effects.

Other considerable cases in biology refer to unique events such as evolutionary facts and mutations, having contextual and general diffusive effects (from dominant to unique), and of an indefinite time duration.

On the one hand, one can distinguish between unique events without embedded or connected anticipatory aspects of their constitutive process, even without cognitive justification available to us for their sudden occurrence. We could say without history, but only understood as temporal sequences, as facts of arbitrary discontinuity. On the other hand, events for which aspects of continuity, compatibility, admissibility and even predictability are recognizable (with reference to previous configurations, and based on our knowledge, such as the historicity of ecosystems), and today form the context of unique and often destructive human interventions.

As it is well known, science has a large variety of approaches and tools available, which are both anticipatory and suitable to model the past.

Turning now to consider the impact of uniqueness on social systems, one can consider unique events such as cases, facts of civilization such as the introduction of writing, music, scientific and technological results, e.g., discovery of fire and invention of the wheel, discovery of vaccines and medicines—all of which can be considered as the result of acquisition and elaboration processes of various types of knowledge, even if highly unpredictable. The diffusion then takes place according to social dynamics of various kinds.

Another case of uniqueness for social systems is the breaking of religions into the temporal evolutionary sequences of human populations, such as the introduction of Buddhism by Siddhārtha Gautama (VI, V century BCE), the introduction of revelations such as the Torah (c. 600 BCE), the preaching of great figures such as Jesus and Muhammad (c. 570 CE). These are considerable events, having high uniqueness, discontinuity with limited compatibility and predictability, or (alternatively, depending on the level of description) as having significant contextual adequacy and compatibility when the context is assumed to act as *incubator*—assumed, anyhow, to have followed increasing spatial diffusion in effects over time.

The uniqueness of these types of events lies in the fact that they are assumed to initiate socially pervasive and diffusive processes, such as conversions and affiliations.

Usually, this type of uniqueness is considered within cultural, conceptual, and philosophical frameworks not caring in the least about the antecedent, focusing only on the power of initiating new processes, new stories. This is obvious for uniqueness achieved after accumulative learning and processing of knowledge and discoveries. However, it is not so obvious when considering events which claim to have overwhelming effects for humanity, such as spiritual salvation of the soul and the prospect of eternal life. It is a matter of presumed salvific and specific, redemptive interventions, having in reality general but subsequent possible effects, such as those mentioned above. The article is about the naïve inconsistency of the understanding of alleged salvific messages without considering the need to consider them in conceptual, philosophical, cultural frameworks where their locality in time and space can be overcome. However, the article is not about religions and theological issues. Given salvific events and visions of salvation that involve or are compatible with exclusion, the article is about the philosophical and cultural understandings of consequential exclusions such as humans as antecedent to some biological evolutionary phases (salvation only for the homo sapiens sapiens?), animals, and other

forms of life. The problem is the consequential consideration of the *excluded as inferior, or are they inferior because they are excluded*? The article is about the social and cultural acceptance of the *inconsistent coexistence* of *salvation and exclusion* in social cultures and usual ways of thinking. This is not intended here as a theological issue as explicitly treated by some religions, but, rather, as an issue that if philosophically unsolved, not debated, and accepted for granted, is compatible with and induces the homo-centrism illusion, egoism, no respect for Nature, and supremacist attitudes. Such issues should be interdisciplinarily elaborated, theologically induced and oriented, to ensure various levels of compatibility.

This is not a critique of religions and of philosophical or humanistic visions of any kind, but has the purpose of soliciting adequate philosophical reflections on a theme which, if neglected and unaddressed, implicitly leaves room for validations of selfish assumptions and homo-centrism: materializing, then, in the assumption of supremacist positions, not based on a respect for Nature and alternative cultural, social, and spiritual visions.

It is a criticism of the fact that in the naïve understanding of the world, like the one in use at the time of the salvific events, people do not ask themselves the problem, implicitly accepting as a matter of fact the exclusion of the rest of non-involved humanity, such as for the lack of food and medicine. This in an assumed, dutiful attitude of subjection and inferiority, in reality understandable as not adequate, *not worthy of the saving messages received*.

The purpose is to underline how cultural and philosophical frameworks of social systems, receiving such messages within the framework of their religions, accept inconsistencies and contradictoriness, such as the selective conditionality of spatial and temporal locality. This conditionality may be solved at the theological level (as in Christianity), but it is a sign of the inadequacy of the cultural and philosophical structures used to interpret these messages. These structures that do not care for the excluded reproduce then this inadequacy in phenomena of egocentricity, selfishness, and homo-centrism in the relationship with Nature and between societies with different levels of well-being, all accepted as phenomenological facts.

We conclude by stressing the need to update these philosophical and conceptual structures so that these alleged inconsistencies and contradictions detectable using ancient and naive conceptions of the world, may be not only theologically solved but also conceptually overcome with more complex understandings of the world, such as considering relativistic time, quantum entanglement, and theories of the universe. This requires theological, philosophical, and scientific interdisciplinarity.

## 2. Maturity of Time

The concept of maturity of time (incubation that has come to maturity, allowing the reception of something previously unrecognizable—something culturally, socially, inadmissible) is often used to indicate appropriateness and adequacy of social evolution; time is needed for such events to happen and to be fruitful (for example, problems and desires appropriate to human ages). In a sort of metaphorical correspondence, the aforementioned interventions have occurred in the maturity of the times: that is, when they become adequate, admissible, understandable, with the possibility of having effects and influencing ongoing and future processes. It is believed that the past would have been inadequate to receive such events, thus implicitly validating the exclusion of all those who found themselves living in it.

The subject may relate to the still philosophical and scientific open question: *what is matter?* What is its changing evolution over time, and its connected space-time-void (its understanding by quantum physics is different from lack of matter)? What is living and non-living matter, mind and matter, biological matter, dark matter, inert matter, antimatter and so on? Some related philosophical aspects are considered in [1].

The concept of maturity of time should mitigate, if not explain, the specificity of time and place of the salvific—however *conditional*—interventions. The specificity of time and space constitute a watershed point between those ('obviously' human beings, but starting

from what evolutionary level?) who have had the temporal and spatial opportunity to at least *consider* it, and those living creatures who have not had this possibility.

Events must, however, happen over time. They have to happen somehow. However, this contrasts with their possible, alleged (or presumed general), *timeless*, salvific effects. Homo-centrism and egoism lead us to the possibility of considering ourselves fortunate, which is in stark contrast to the unconditionality of salvific messages themselves. *This inconsistency, rather than understood as human presumptuous implicit reproach, should actually be understandable as a* poor *understanding of the saving messages, themselves.*

This does not relate to the salvific messages themselves, as considered by religions, but, rather, to the cultural and philosophical frameworks used to formulate and understand them. The focus is on the fact that the usual understandings of the salvific messages are based on concepts of pre- or classic physics dealing with naïve concepts of time, space, and the universe. This allows for distortions which validate approaches compatible with or even implying elementary linear homo-centrism and accepting conditional spatial and temporal localities.

Only theologically, as a truth of faith, Christ, for example, can be considered the *Salvator Mundi* (the title of a painting attributed to Leonardo), and not just the eventual savior only of human beings who lived after Christ and were and are recipients of the salvific message. As in the catechism of the Catholic Church, Part 1, Section 2, Chapter 2, Article 5, Paragraph 1:

> [634] The descent into hell brings the Gospel message of salvation to complete fulfilment. This is the last phase of Jesus' messianic mission, a phase which is condensed in time but vast in its real significance: the spread of Christ's redemptive work to all men of all times and all places, for all who are saved have been made sharers in the redemption.

The same is true for the Torah (see below) and other salvific events in other religions.

Is there a supposed level below which, e.g., of a biological, cognitive nature, the salvific intervention does not apply, or have no effects?

Can the inconsistency considered be resolved only theologically? This article intends to introduce the possibility of identifying philosophical and scientific conceptions having compatibility with resolutions, explications of events of this nature, making them conceivable and admissible in a modern conception of the world. The article is about the need to philosophically and interdisciplinarily elaborate the inadmissibility of exclusion. The point is the admissibility of the exclusion and how to assume philosophical, cultural, interdisciplinary approaches intrinsically incompatible. Such ethical inadmissibility should be conceived philosophically and interdisciplinarily.

As already considered by Galileo, theological truth does not necessarily coincide with the scientific and philosophical one. Indeed, it is a question of admitting the separability of salvific events and messages from the conceptions of the world in use when they occurred and with which they were considered and described. The fact of creating and devising conceptual, scientific, and philosophical contexts compatible—even if not necessarily explanatory or alternative—with such events does not have the purpose of reducing their religious and theological significance, but of allowing understandings without consequential homo-centrism and selfishness admitting exclusion. Without this adjustment, the unique salvific events can affect only religiously, that is, optionally, on the scientific and philosophical conceptions used. Without this adjustment the homo-centrist illusion and egoism prevail as the natural, linear consequential attitude and these questions are not raised. It rather assumes that the intervention is for involved human beings, and moreover for those who have had and are lucky enough to receive and practice it, ignoring the enormous inconsistency within the message itself, and its presumed general and available salvific nature.

The inconsistency should be declared and explicitly faced when considered in the original ancient scientific and cultural frameworks, to avoid the hypocrisy of ignoring it. We believe that, more than a problem, it is an opportunity to extend the grandeur of such

unique events by introducing suitable understandings in new scientific and philosophical frameworks based, for instance, on new understandings of space and time, general long-range networking and interdependence, and new cosmological understandings of the universe. In these contexts, homo-centrist illusion and egoism, non-respect for Nature, even if still applied, appear at least *theoretically* unsustainable and wrong because of their elementary naiveness, lack of correlation, lack of general and long-range interdependence.

Another possibility is to understand the salvific messages *as constituted of multiple uniqueness*, at least within the current realm of human understanding. This multiplicity should conceptually correspond to how science considers both different, non-equivalent aspects of same phenomena, such as: (a) thermodynamic, mechanical, electromagnetic, and optical phenomena in physics; and (b) physiological, biochemical, physical, psychological, social, hygienic, alimentary, environmental, and stress-related phenomena in medicine. However, such multiplicity does not only relate to aspects, but it is even constitutive of and intrinsic to multiple phenomena, such as in collective behaviors, where elements perform multiple roles, or in multiple systems, where the same components can play interchangeable, multiple, and overlapping roles. For instance, in electronic devices, values adopted by the same components have simultaneous, multiple meanings, e.g., they are simultaneously related to safety control and to regulation, or both to programs and to nodes of the Internet, simultaneously allowing user geolocation and profiling. *Such an issue is fundamental in physics when considering the uncertainty principles, such as the simultaneous corpuscular and wave natures of particles in physics* [2] (pp. 42–45, 166–170), [3].

This understanding is compatible, for instance, with religions such as Hinduism (c. 500–200 BCE), a real multiple religion. Actually, Hinduism is a system of thought, concepts, cosmological visions, metaphysics, mythology, philosophies, pilgrimage sites, rituals, and texts. Even in Christianity, Christ is intended as God and human being. This is completely different from paganism, or naive interpretations of separate multiplicities.

The maturity of time could refer to the adequacy of the time in which such events are introduced, but also to the fact that there is an implicit, mature need for interventions of this type and therefore also for their emergence and self-constitution into a state capable of establishing necessary properties, such as intelligence and spiritual sensitivities. We remind readers that, in the science of complexity and (very briefly), in the populations of interacting elements of any kind, self-organization can be understood as the occurrence of prevailing (and sustained) synchronizations, be they singular, multiple, or partial. Emergence can be understood as the prevailing dynamic of multiple, incomplete, temporal, and overlapping phenomena of self-organizations, leading to *dynamical coherences*. Social behaviors are emergent, e.g., in the establishment of flocks, swarms, markets, and consensus [4,5].

The social necessity, the cultural need for certain religiosity and cultural visions, is envisaged also as a response to existential issues and social practices. This contrasts with revelatory views, and is a matter for study by the sociology of religions. See, for instance, [6,7] and the history of religions; see also, for instance [8].

However, we may consider coexistence of the two views. In the same way as we consider practical coexistence between *objectivism* and *constructivism.* The former assumes a world naively intended as fixed, only to be discovered—in contrast with the multiplicity in physics mentioned above—whereas in the latter, experiments are considered as questions put to Nature, which answers by making events happen. No question means no answers, while corresponding events may be considered as answers to proper questions to be invented by the researcher. Even these two approaches, i.e., revelations and emergent understandings, could very well coexist and support each other, up to the point of requiring episodes of mutual incompatibility as extreme cases. This theme is combined with the constructivist truth, which excludes uniqueness and static nature as (at best) unrealistic in science [9] (p. 6), [10].

Conceptually, the case of *the excluded* is considered here: that is, those who lived prior to, or have not, for various reasons, been involved in events as such, because of physical, linguistic, cultural, or intraspecies/biological distance (a situation not even considered by

homo-centrism foregone). We conceptually consider the unfortunate who have lived out of the maturity of time, or who have not been and are not affected by such unique events and their effects. Are such events interesting only human beings at a suitable level of evolution?

It is a theme whose ignorance, often guiltily taken for granted and not explicitly considered, culpably reduces the philosophical, ethical, and conceptual consistency of the single events with their generalized claims, such as those salvific and universal in scope.

Such neglect is compounded on the part of those who are presumably reached, by selfishly elaborating messages of salvation *without looking back and around*.

This attitude gives rise to implicit, undeclared contradictions, emerging in various other forms of inconsistency and selfishness, such as homo-centricity towards Nature.

The philosophical understanding and interpretation of such messages should be properly reconsidered.

The philosophical basis assumed compatible with such neglection, is fragile in terms of inconsistency, where many issues are neither declared nor addressed. These are philosophical problems, but are also, in some cases, understandable as theological: e.g., is it the Jewish Tiqqun 'olam, saving the world from now on? Concerning only those involved, in various capacities? That is, in the same way as some people were metaphorically lucky to be able to drink an antidote: only for humans?

It should also be noted that such a conceptual and philosophical context assumes the view of classic physics, neither relativistic nor quantum, with a unidirectional, naïve conception of time (see below).

The situation is graphically illustrated in the painting reproduced in Figure 1, where, probably for the first time, the perspective of the crucifixion of Christ from behind the cross is considered. The centrality of Christ is maintained, but not in a self-centered, homo-centric way, and is addressed to the present, as a promise of the future (in a classic, unidirectional understanding of time) only. Some of those excluded are metaphorically represented. Pierino Foscarini painted the picture in 2022, on the advice and inspiration of the author and, it is located in the author's private home.

In place of Jesus, there may be the back of Moses presenting the Torah tablets to the Jews, or the back of the prophets and the revelators. It is also possible to discern those Jews who have *not* succeeded in joining the multitude participating in the passage of the Red Sea, and are only watching them (see below).

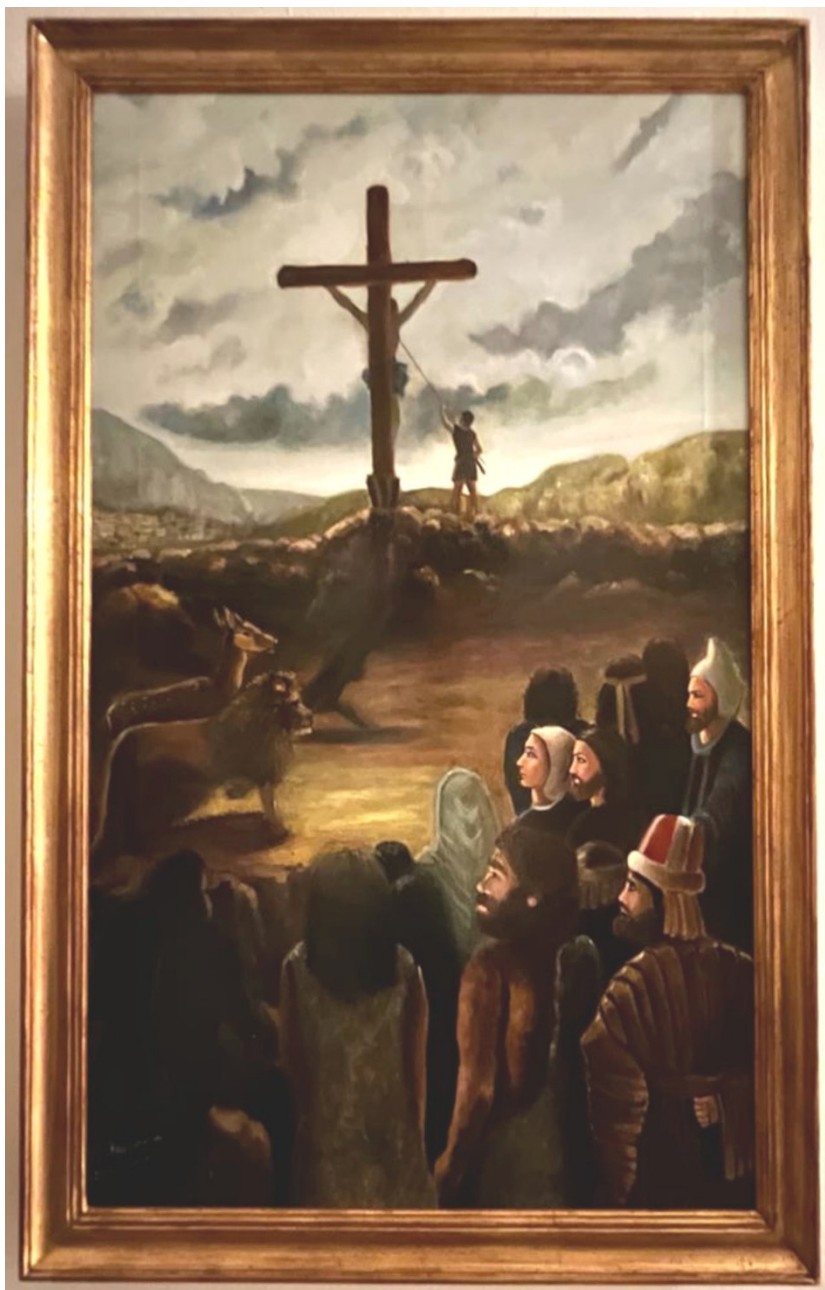

**Figure 1.** The excluded.

### 3. The Painting

The scene represented is considered as a theme for philosophical discussions relating to the understanding of socially unique events and the role of the excluded, as mentioned above.

In the painting, we understand that it is the Christ because there is the soldier who pierces him in the side, as told by the Gospels. The numerous classic representations of the crucified Christ consider the audience in front, who are interested in the scene at various levels, from the followers, to the soldiers, to the Jewish religious present, and others whose level of interest is not known (for instance intending it as a crucifixion, like others, frequently used by the Romans).The world depicted behind the cross represents, metaphorically, others not only at that moment, but also in a timeless context, in which space and natural relationships are suspended. The subject of the excluded is metaphorically represented by the *invisible* (in time and space) audience, behind the crucifixion considered

by Christianity. The remainder of this article considers how equivalent situations may be represented in other religions.

This salvific instant is intended to be timeless.

Let us consider some details in particular.

- The lion and its typical prey are side by side, as if stunned to the point of assuming unnatural behavior: the lion does not hunt and the prey does not flee. It can be assumed that these scenes happen during the crucifixion of Jesus, but in other, unspecified places. They are only amazed by the grandeur of the event, all the more because it is unknown and remote, which generates a general unnaturalness. There is an incomprehensible light in front of them, from an unspecified source. The grandeur of salvation astounds; it momentarily breaks time and the rules of Nature, like lightning in the *whole* world, simultaneously. But, coming out of this context, it will not change their roles in Nature, in biological life, which will then resume as if nothing had happened. However, this has consequences for the entire known world at the time, excluding, for instance, the Americas, Asian countries, Oceania, and the universe(s)—not to mention possible alien life forms that we would have difficulty even recognizing.

- Some people timelessly, side by side, look at the back of the Cross. The entire past is concentrated in an instant. They are also stunned, maybe disappointed, excluded. Primitive man is unable to stare at even the back of the cross as well as priests and personalities of previous developed societies (such as Phoenicians, Carthaginians, earlier Romans, Greeks, Babylonians, and Mayans), who are likewise excluded and amazed. Aristotle, Euclid, and Plato could be among them. Or, perhaps they are not excluded—or they are excluded only if we mean the message as necessary only for the future. The light in front is incomprehensible. The light does not distinguish the dark to illuminate, it is just available. It is there. Our eyes see what we can see, we understand what we can understand. A possible philosophical, theological question perhaps may focus also on the hypothetical idea that *the excluded do not need to be saved*. *The maturity of time could be then since when salvation is necessary.*

The focus of the article is on the role of the excluded, theologically considered in some religions such as Christianity, and requiring philosophical and scientific conceptions suitable to give them significance rather than egoistic neglect and non-consideration.

## 4. The Philosophical Nature of Exclusion

The theme here concerns inconsistencies and superficiality in the narratives and treatments of the unique salvific events, without any reference to those events' possible effects upon the past, or on contexts of a different nature, such as different kinds of (i.e., nonhuman) living beings. The case is represented in the painting and when considering, for example, those Jews who, for practical reasons (see below) have *not* succeeded in participating in the passage of the Red Sea (see below).

Theological interpretations are capable of resolving the inconsistency. However, they are considered here as to be accompanied by compatible philosophical and scientific reformulations, not necessarily substitutive, in order to resolve contradictions, in a suitable conception of the world. Leaving the solution of the problem only to theology while maintaining the use of concepts admitting and compatible with exclusion inevitably validates homo-centric and selfish conceptions.

This is not to replace the theological interpretation, but to consider it as a forerunner of reformulations of the conception of the world at least intrinsically incompatible with admitting exclusions and selfishness, that is, homo-centrism and non-consideration for Nature.

The subject should be debated by taking into account the modern understanding of space and time, a quantum understanding of the world, and corresponding theories of the universe. The events and sacred books are usually presented and discussed by assuming as valid their implicit understanding of the world. Galileo introduced the possibility of metaphorically understanding the physical concepts of sacred texts. However, their

consideration and treatment in a world conceived differently would not be understood philosophically as a violation, but as an adaptation (the sacred texts are, after all, translated; why not adapt the world where they are considered to have taken place?). As such, they are also an adaptation aimed at overcoming conceptual inconsistencies and unexplained determinisms, as exemplified below. Another possibility is to *explicitly* consider the inconsistency discussed above as a mystery: that is, not rationally solvable, nor understandable by human beings, but philosophically processable as similar cases, e.g., death and life, evil, pain, and their potential consequences. Moreover, in mathematics we use incomputable imaginary numbers [11]. Furthermore, *the maturity of time may be intended as degeneration, and not as improvement, reaching a state from whence salvation is necessary.*

Remaining within the context of Christianity considered in the painting, there are roles whose inalienability and inescapability have been historically, socially, and philosophically ignored.

The crucifixion seems to be in front of a stage, full of inevitable roles. We can ask ourselves, how Jesus should have died instead: from disease, by accident, being torn to pieces, from hunger, or from old age? Of course not. In that case, his death would have been irrelevant and certainly not celebrated. Instead, it was the celebration of a holocaust. The Roman soldiers who crucify him, Pilate, the Jewish leaders who condemn him, all play irreplaceable roles—but without blame?

In reality, they are *officiating without knowing it.* They are all priests of a unique event, as well as the prophets of the Torah and the Jews who built the golden calf. This is combined with the adequacy of the event within the context of its occurrence, and which allows the event to have unique global effects. Otherwise, ignoring the context, its uniqueness would not even be recognized and would be extinguished by insignificance.

One might wonder *when we are* not *officiants, at least as we are witnesses of unique events,* such as births, deaths, accidents, discoveries, encounters and separations, natural events, and many others that remain in the memory?

There are countless ways to play a part, all of which are highly equivalent. But what matters is the part, the role, not the player, to be understood as a mix of constructivism and objectivism, of determinism and freedom.

For instance, in the in the Gospel of Saint John (13:21–30) it is written [12]:

> [21] When he had said this, Jesus was deeply troubled and testified, "Amen, amen, I say to you, one of you will betray me". [22] The disciples looked at one another, at a loss as to whom he meant. [23] One of his disciples, the one whom Jesus loved, was reclining at Jesus' side. [24] So Simon Peter nodded to him to find out whom he meant.[25] He leaned back against Jesus' chest and said to him, "Master, who is it"? [26] Jesus answered, "It is the one to whom I hand the morsel after I have dipped it". So he dipped the morsel and [took it and] handed it to Judas, son of Simon the Iscariot. [27] After he took the morsel, Satan entered him. So Jesus said to him, "What you are going to do, do quickly". [28] [Now] none of those reclining at table realized why he said this to him.

It appears that Judas had such a *necessary* role. As in nature, for example, the roles of prey and predators.

And again, when in the Torah [13], in the book of Exodus, 12 it is written [14]:

> [11] When Pharaoh saw that there was relief, he hardened his heart, and he did not hearken to them, as the Lord had spoken.

The Pharaoh was a necessary celebrant.

And in many other cases: countless ways to play a part.

These observations are introduced not to reduce unique events to mechanical determinism, but to highlight how unique events such as the story of Jesus could only happen that way or equivalent in our world, assuming approaches based on classic physics and its naïve conception of time.

In our daily world, experienced and understood as classic, we see uniqueness, equivalences, determinism, irreversible temporal directionality, and unrepeatability that can each be understood in a completely different way when we abandon the classical vision, the only one available at the time of the unique events mentioned. *It is now not only possible, but culturally and ethically* necessary *to reconsider such events in new conceptual contexts—those based, for instance, on concepts formulated by non-classic theoretical physics, on new philosophical, anthropological, and natural science concepts.*

Back to roles, it is interesting to consider how the theme of exclusion with regard to the presumed *exclusive centrality* of the Jewish people (intended instead as a context of adequacy, compatibility, and responsibility, incubator and stage) recalls what is written in the Torah [13], in the book of Exodus 12 [14]:

> [37] The children of Israel journeyed from Rameses to Succoth, about six hundred thousand on foot, the men, besides the young children.

> [38] And also, a great mixed multitude went up with them, and flocks and cattle, very much livestock.

From the text we understand that not only the Jews left, but also many foreign people, presumably those who had reasons to run away, as well as *small and large animals, extremely numerous cattle*. And the Promised Land was also for them: it was enough to have decided to leave with the Jews [15]. They too will eat manna.

The excluded are those who, for some reasons, did not know they were leaving, or who could not leave, having children—or the elderly, who were sick, or prisoners, who were afraid. It reminds us of the excluded in the painting.

What kind of salvation does it exclude, for practical purposes? Is salvation an *opportunity* to be seized, as the thief crucified with Jesus seems to have done? It seems that the figures in the painting lost the opportunity. *Is the message that we have to be quick, to take opportunity on the fly?* Here, we want not to judge, but to consider the coherence with the entire salvatory message, assumed unconditional: the general philosophical, ethical, and morally conveyed message. We must understand well beyond appearances and make conceptions of the world almost compatible, even if not necessarily explanatory or alternative, with theological understanding.

A possible rule of exclusion is outlined, in principle, as when in the Gospel Mathew 13:9, which reads: "Whom ever has will be given more, and he will have an abundance. Who ever does not have, even what he has, will be taken from him".

This topic has been presented in a lecture by Arne Collen and the author at the Temple Isaiah, Lafayette, CA (Rabbi Graetz), considering the conceptual correspondence with first-order phase transitions in physics [16]:

> "I want to give a very simple example in physics. In physics we have the so-called phase transitions. An example is the transformation of water into ice. When we reach a specific temperature, the process occurs. The process does not occur in a yes or no way. When the temperature is very close to zero, the water is becoming very cold. When the temperature decreases very slowly it is possible to have the water still in a liquid phase even at temperature less than zero. In physics we call this a meta-stable condition, when it is sufficient to reduce the temperature very little and pow, you have the transformation into ice. So the metaphoric idea is that if in your personal story in life you have the ability to reduce the temperature and you reduced sufficiently to reach the critical point for the phase transition, then you have the phase transition occurring. But if in your life, you were able to reach any low temperature getting close to the phase transition point, but at the very last moment when control is lost, the critical point is not reached, then you will start to reverse the process and the temperature will increase leaving far from the possibility for the phase transition to occur".

It is necessary to interpret from the perspective of a general, unconditional salvation.

In this regard, we like to mention Von Foerster, father of Cybernetics, who considered that there are no *anomalies* in the environment. Rather, in such a case, it is a matter of inappropriate approaches, concepts, and models employed to understand the phenomenon under consideration [17]. It is appropriate to consider the anomaly of the inconsistency—i.e., anomaly between salvatory unique events and their temporality and localization, excluding generations of human beings and all other living beings—by using an abductive approach [18].

The painting wants to celebrate the need to elaborate, to consider the unconditionality of salvatory messages, and also to avoid the inconsistencies that we find ourselves continuously emerging from, in various forms.

## 5. Conclusions

We mention above the inconsistency between unique, general, long-range salvific messages and their temporal and spatial localization, implying the impossibility of retroactive and global validity. Unique salvific messages seem to have the characteristic of local opportunities, which are then (possibly) diffusible. How could it be otherwise? In this regard we considered such understanding as related to concepts of classical physics, naively adopted, in the texts considered sacred. The concepts presented in sacred texts are even more antecedent of the classic physics. The theme concerns inconsistencies and superficiality in the narratives and treatment of the unique salvific events, without any reference to the possible general and retroactive effects, such as those represented in the painting in Figure 1. The search for possible solutions is left to theology.

Must we necessarily assume the physical conceptions (for example, the spatial and temporal experiences) that the protagonists of the salvific and sacred events had? The only examples of conceivable violations were the possible prophetic predictions, the prophecies. For example, in the Torah, the bush that burns without consumption, the water that comes out from the rock, the manna, and, in the Gospels, Jesus walking on the water, the multiplication of the loaves and fishes, and the resurrection of Lazzaro.

We comment that the subject should be debated by taking into account the modern scientific and philosophical understanding of the world looking for representations whose possible compatibility with the theological solutions to make it possible to implicitly overcome the admissibility of the excluded. New approaches should be considered, such as a new relativistic understanding of space and time, general networked systemic interdependence, phenomena of acquisitions of new properties such as in phase transitions and emergence, a quantum understanding of the entangled world. Consider, for instance, *quantum retrocausation* [19,20] and other forms of so-called *reverse causality*, and other theories of the universe. There are varied approaches to providing an interpretation of the predictive and explanatory success of quantum theory. One class of interpretations hypothesizes backward-in-time causal influences, i.e., retrocausality. Speculatively, physicists consider that if there is retrocausality in the universe, it might have occurred in certain eras, perhaps close to the Big Bang. "When and where" are not a definite arrow of causality, as we have established. Furthermore, in several equations across quantum mechanics and general relativity, physicists consider *imaginary time*, a concept intended as a mathematical simplification of time.

In this regard, we mention the Big Bang Theory, introduced in 1927 by the Belgian Catholic priest Georges Lemaître [21]. This theory, in short, says that the universe started with a small *singularity* and became the cosmos, inflating over the next 13.8 billion years. Lemaître says, " . . . if we go back in the course of time we must find fewer and fewer quanta, until we find all the energy of the universe packed in a few or even in a unique quantum". It is a matter of accounting for different conceptions of the universe in cosmology. Will salvatory sacred messages survive and even extend when intended in different time and space, conceptual contexts, and universes?

This should be a matter of responsible philosophical and theological interdisciplinary debate, involving science that is usually excluded, if not considered incompatible [14].

In a context of interdisciplinarity, assumed to occur when problems and solutions in one discipline can be used in another, concepts can be mutually defined and correspond, and the entities being considered may have multiple meanings.

It is an interesting terrain of interaction between science, theology, and the philosophy of religion, perhaps a terrain necessary to overcome, in its centuries-old omission by philosophers and scientists. Perhaps, too, to overcome the terrain which contributed to the emergence and validation of homo-centric visions, conflictual and political philosophical religiosities.

This is what the painting wants to remind us.

**Funding:** This research received no external funding.

**Conflicts of Interest:** The author declares no conflict of interest.

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
