# Peer review of "The Role of the Excluded"

_philosophies, doi:10.3390/philosophies7040083_

Round 1
Reviewer 1 Report
The paper is clearly written and has the ambitious goal of focusing on a strange topic – the salvific message brought forward by different religions – from different perspectives. Of course, adopting different perspectives to address this particular topic should increase the complexity of the arguments and the references to the appropriate literature. Unfortunately, in this particular case, it didn’t have this effect. Thus, in order to improve the quality and the reasonableness of this paper, I would encourage the authors to rewrite it acknowledging and engaging with the appropriate literature, and maybe also discussing other potential solutions to the inconsistency that is the key topic of this essay.
To proceed in order, I should say that the inconsistency this paper focuses on is the one that emerges when considering the salvific message of religions as universally applicable (so touching also people that could not have heard the message because spatially distant or temporally precedent to this message). The authors claim that this inconsistency should be a topic of philosophical, theological, and broadly cultural debate, and should be approached by considering the updated ideas of time and matter in contemporary physics. Thus, the authors assume that there is no literature to read regarding the topic of salvation and theological mystery, time and the concept of God, and the inconsistencies between universal salvation, theological redemption, and original sin. Unfortunately, a simple and quick search on google scholar would have provided the authors the means to adjust this view: there is in fact a huge amount of theological and philosophical literature on the core topic of this paper. To my knowledge, the only original take on this theme that has been provided in this paper is the potential update of the concept of time and matter according to current physical theories. At the same time, this original take loses importance and reasonableness since the authors do not discuss and engage with the philosophical and theological literature that actually deals with the topics of salvation, theological mystery, and time (not to mention aesthetic and artistic discussions on the painting the authors explicitly cite and discuss). I’m not saying that the authors should prove that there is a lack in the literature regarding this theme: as it is known, it’s philosophically and scientifically troubling to prove the existence of an absence. At the same time, there is a fundamental lack of references to philosophical and theological works in this paper, which is hard to deny if we look at the references list on the last page of the article. If the authors are convinced this topic is not discussed in the theological and philosophical literature (which is, in part, simply not true), they should at least cite their theological and philosophical sources to make their argument convincing. This is the main problem (and the most troubling one) of this essay and should be overcome to make this work publishable in any philosophical journal.
There is also another issue that concerns the writings that is surely connected to the main problem but should be addressed also apart from it: there are different sentences in italics that the authors put forward as a reminder of the goals of the paper and its alleged novelty. Examples of these sentences are:
“This should be matter of philosophical debate. (p. 3, but also 7 and 10)”
“This attitude gives rise to implicit, undeclared contradictions, emerging in various other forms of inconsistency and selfishness, such as homo-centricity towards Nature and complicity with facts of political and military power.”
“We wish to highlight the philosophical inconsistency of the neglections mentioned above.”
“This salvific instant is intended to crosses all times and the entire world.”
“Will salvatory sacred messages survive and even extend—for example, in consistency—when intended in different physical, conceptual contexts and universes?”
Most of these sentences are not completely understandable in the text: they are presented as matter-of-fact sentences, but they should be defended (again engaging with the appropriate literature) since they represent the key arguments of this paper. In this form, these sentences represent just-so arguments that cannot be taken seriously in philosophical debates.
Author Response
Dear reviewer,
Thank you for your careful reading and comments.
They were very helpful to realize how unclear the general topic of the article was and I apologize for that.
The article is not about religions and has not theological issues.
This is why there are not theological works in the references.
The article is about the naïve inconsistency of the understanding of alleged religious salvific messages without considering the need to consider them in suitable conceptual, philosophical, cultural frameworks, even if not necessarily explanatory or alternative with theological understanding, where their locality in time and space can be overcome. At the moment this inconsistency is considered solved by theology in some religions, as in Christianity:
634 "The gospel was preached even to the dead."483 The descent into hell brings the Gospel message of salvation to complete fulfilment. https://www.vatican.va/archive/ENG0015/__P1R.HTM#$S7
I lacked to make evident that the inconsistency lies in the simplified, ancient, linear understanding. A more appropriate understanding is required as Von Foerster introduced for systems science (p. 9). The adoption of understandings ignoring such inconsistency and its unethical selfishness is claimed having subsequent validation of homocentric and egoistic approaches.
I ask if the inconsistency considered can be resolved only theologically. This article in-tends to introduce the possibility of identifying philosophical and scientific conceptions suitable to resolve and explain events of this nature, making their solutions conceivable and admissible in a modern conception of the world.
The fact of creating and devising conceptual, scientific and philosophical contexts compatible -even if not necessarily explanatory or alternative of theological understandings- with explications and solutions of such events does not have the purpose of reducing their religious and theological significance, but of allowing understandings having not consequential homo-centrism and selfishness admitting exclusion. Without this adjustment, the unique saving events can affect only religiously, that is, optionally, on the scientific and philosophical conceptions used. Without this adjustment the homo-centrist illusion and egoism prevail as natural, linear consequential attitude and does not raise these questions.
Indeed, it is a question of admitting the separability of salvific events and messages from the conceptions of the world in use when they occurred and with which they were considered and described.
I added new text.
_____________
Yes, an “original take on this theme that has been provided in this paper is the potential update of the concept of time and matter according to current physical theories”. I think it is not negligible. However, a point is “Will salvatory sacred messages survive and even extend when intended in different time and space, conceptual contexts and universes?” and retroactive causation (p.10)? The focus is on the need to understand theological solutions in new almost compatible philosophical and scientific frameworks having not consequential acceptance of the exclusion (p.11).
_____________
“Thus, the authors assume that there is no literature to read regarding the topic of salvation and theological mystery, time and the concept of God, and the inconsistencies between universal salvation, theological redemption, and original sin.”
I do not make this assumption. The focus is on the fact that the usual understandings of the salvific messages are based on concepts of pre- or classic physics dealing with naïve concepts of time, space, and universe. This allows distortions validating approaches having consequential homo-centrism and accepting conditional spatial and temporal localities.
Please note the several consequent changes and updates introduced.
_____________
As regards:
“This should be matter of philosophical debate. (p. 3, but also 7 and 10)”
p.3-removed
p.7-removed
p.10-Italics removed
____________
“This attitude gives rise to implicit, undeclared contradictions, emerging in various other forms of inconsistency and selfishness, such as homo-centricity towards Nature and complicity with facts of political and military power.”
p.3-Revised
____________
“We wish to highlight the philosophical inconsistency of the neglections mentioned above.”
p.5 removed
____________
“This salvific instant is intended to crosses all times and the entire world.”
p.7 revised
____________
“Will salvatory sacred messages survive and even extend—for example, in consistency—when intended in different physical, conceptual contexts and universes?”
- 10-Revised
____________
Thank you for comments.

Reviewer 2 Report
We agree with the author that the need for an interdisciplinary approach is great and that a dialogue between science and religion is needed. But the main thesis of the article – »inconsistency between unique, general, long-range salvific messages and their temporal and spatial localization, implying the impossibility of retroactive and global validity« is something we find very problematic because it addresses the theological problems, which ca nor cannot be answered within a particular religious tradition. Since the critique is directed primarily at Christianity, we are surprised by the author’s ignorance of Christian doctrine, which contains answers to some of the theological questions raised by the author. Some of them are located in the Catechism of the Catholic Church, which is available online.
Regarding the question of »retroactive validity« of the Christian message, we can find the answer in paragraphs 632-635:
Source: https://www.vatican.va/archive/ENG0015/__P1R.HTM
632 The frequent New Testament affirmations that Jesus was "raised from the dead" presuppose that the crucified one sojourned in the realm of the dead prior to his resurrection.477 This was the first meaning given in the apostolic preaching to Christ's descent into hell: that Jesus, like all men, experienced death and in his soul joined the others in the realm of the dead. But he descended there as Saviour, proclaiming the Good News to the spirits imprisoned there.478
633 Scripture calls the abode of the dead, to which the dead Christ went down, "hell" - Sheol in Hebrew or Hades in Greek - because those who are there are deprived of the vision of God.479 Such is the case for all the dead, whether evil or righteous, while they await the Redeemer: which does not mean that their lot is identical, as Jesus shows through the parable of the poor man Lazarus who was received into "Abraham's bosom":480 "It is precisely these holy souls, who awaited their Saviour in Abraham's bosom, whom Christ the Lord delivered when he descended into hell."481 Jesus did not descend into hell to deliver the damned, nor to destroy the hell of damnation, but to free the just who had gone before him.482
634 "The gospel was preached even to the dead."483 The descent into hell brings the Gospel message of salvation to complete fulfillment. This is the last phase of Jesus' messianic mission, a phase which is condensed in time but vast in its real significance: the spread of Christ's redemptive work to all men of all times and all places, for all who are saved have been made sharers in the redemption.
635 Christ went down into the depths of death so that "the dead will hear the voice of the Son of God, and those who hear will live."484 Jesus, "the Author of life", by dying destroyed "him who has the power of death, that is, the devil, and [delivered] all those who through fear of death were subject to lifelong bondage."485 Henceforth the risen Christ holds "the keys of Death and Hades", so that "at the name of Jesus every knee should bow, in heaven and on earth and under the earth."
Regarding the question of »global validity« of Christian message i.e. "Outside the Church there is no salvation?" we can find the answer in these paragraphs:
Source: https://www.vatican.va/archive/ENG0015/__P29.HTM
§846 How are we to understand this affirmation, often repeated by the Church Fathers? Re-formulated positively, it means that all salvation comes from Christ the Head through the Church which is his Body:
Basing itself on Scripture and Tradition, the Council teaches that the Church, a pilgrim now on earth, is necessary for salvation: the one Christ is the mediator and the way of salvation; he is present to us in his body which is the Church. He himself explicitly asserted the necessity of faith and Baptism, and thereby affirmed at the same time the necessity of the Church which men enter through Baptism as through a door. Hence they could not be saved who, knowing that the Catholic Church was founded as necessary by God through Christ, would refuse either to enter it or to remain in it.
§847 This affirmation is not aimed at those who, through no fault of their own, do not know Christ and his Church:
Those who, through no fault of their own, do not know the Gospel of Christ or his Church, but who nevertheless seek God with a sincere heart, and, moved by grace, try in their actions to do his will as they know it through the dictates of their conscience - those too may achieve eternal salvation.
§848 "Although in ways known to himself God can lead those who, through no fault of their own, are ignorant of the Gospel, to that faith without which it is impossible to please him, the Church still has the obligation and also the sacred right to evangelize all men."
The Catechism of the Catholic Church is just one of the Catholic documents addressing the theological problems highlighted by the author. It would be neither right nor academic for the author to criticize a particular tradition without looking at what answers to internal problems and anomalies the tradition has found itself. Every tradition has problems and anomalies, as Alasdair MacIntyre pointed out, but they can be solved to some extent by the tradition itself. What about other religious traditions as far as the issues addressed by the author of the article are concerned? The author should inquire about their theologians, e.g. Islamic theologians.
Author Response
Dear reviewer,
Thank you for your comments.
They were very helpful to realize how unclear the general topic of the article was and I apologize for that.
Thank you for your quotations.
____________
The article is not about religions and has not theological issues.
This is why there are not theological works in the references.
You wrote “…Since the critique is directed primarily at Christianity…”.
First of all, the article considers salvific messages in religions and not only in Christianity, such as the case for Jews, see p. 6 and 9.
Second, the inconsistency mentioned is intended due to inappropriate cultural, philosophical and ancient scientific understandings, see the end of the Introduction p. 3 and more. These not only tolerate inconsistencies, probably theologically solved as in your appreciated citations, but may validate conditionalities and localities used in other contexts allowing validation of homocentric and egoistic approaches towards Nature and between societies with different levels of well-being, accepted as phenomenological facts.
____________
The article is about the naïve inconsistency of the understanding of alleged religious salvific messages without considering to need to consider them in suitable conceptual, philosophical, cultural frameworks, even if not necessarily explanatory or alternative with theological understanding, where their locality in time and space can be overcome. At the moment this inconsistency is considered solved by theology in some religions, as in Christianity:
634 "The gospel was preached even to the dead."483 The descent into hell brings the Gospel message of salvation to complete fulfilment. https://www.vatican.va/archive/ENG0015/__P1R.HTM#$S7
I lacked to make evident that the inconsistency lies in the simplified, ancient, linear understanding of the world. A more appropriate understanding is required as Von Foerster introduced for systems science (p. 9). The adoption of understandings ignoring such inconsistency and its unethical selfishness is claimed having subsequent validation of homocentric and egoistic approaches.
I ask if the inconsistency considered can be resolved only theologically. This article in-tends to introduce the possibility of identifying philosophical and scientific conceptions suitable to resolve and explain events of this nature, making their solutions conceivable and admissible in modern conceptions of the world.
The fact of creating and devising conceptual, scientific and philosophical contexts compatible -even if not necessarily explanatory or alternative of theological understandings- with explications and solutions to the exclusion considered by such events does not have the purpose of reducing their religious and theological significance, but of allowing understandings having not consequential homo-centrism and selfishness admitting exclusion. Without this adjustment, the unique saving events can affect only religiously, that is, optionally, on the scientific and philosophical conceptions used. Without this adjustment the homo-centrist illusion and egoism prevail as natural, linear consequential attitude and does not raise these questions.
Indeed, it is a question of admitting the separability of saving events and messages from the conceptions of the world in use when they occurred and with which they were considered and described.
I added new text.
_____________
I do not assume that there is no literature to read regarding the topic of salvation and theological mystery, time and the concept of God, and the inconsistencies between universal salvation and theological redemption.
In the article the focus is on the fact that the usual understandings of the salvific messages are based on concepts of pre- or classic physics dealing with naïve concepts of time, space, and universe. This allows distortions validating approaches allowing homo-centrism and accepting conditional spatial and temporal localities.
Please note the several consequent changes and updates introduced.
I added new text.
Thank you for comments.

Round 2
Reviewer 1 Report
I thank the author for the changes made. Unfortunately, I believe that, by general knowledge, I cannot understand in which sense “the article is not about religions and has not theological issues”. Indeed, what does “salvation” mean if we only consider its role in social unique events such as “the introduction over time of the Torah, Hinduism, Buddhism, Christianity, and Islam”? These examples have all religious claims in common and among them, the idea of salvation is extremely religious and theological: indeed, in no other meaning, we can say that those events are or claim to be salvific for humanity (in whichever meaning this last word is taken). The inconsistency the author finds is so primarily theological: indeed, if I don’t believe in God and I don’t believe any of those events mean something religious, spiritual, or theological to me – indeed, I only take them to be historical and cultural important phenomena but not salvific. I so don’t think I have been saved by them (I didn’t think I needed to be saved in the first place) so my sense of supremacy (if I have one) does not derive from the salvation which I don’t think I have or want.
I would, of course, change my mind and I would much appreciate if the author could provide a way for the readers to understand the concept of salvation (as brought forward by those religions) that does not involve theological and religious meaning.
It would also improve the overall clarity of the paper if the author could provide a way to understand how the problem of the excluded means that “the homo-centrist illusion and egoism prevail as natural, linear consequential attitude”.
These are still two enormous problems of the paper, which make it unclear and speculative.
Consequently, the problematic absence of theological literature or of even philosophy of religion (for example, Peter Abelard and Søren Kierkegaard are mentioned but there are no bibliographical references on them) still remains a huge third drawback.
Until these problems are not solved I don’t see a way to accept this paper for publication.
Author Response
Dear Reviewer,
Thank you for the comments. Probably I was unclear and thanks for making me to clarify.
Regarding the first and second points I inserted the new text that I hope will solve the problems:
“The article is not about religions and has not theological issues. Given salvific events and visions of salvation that involve or are compatible with exclusion, the article is about the philosophical and cultural understandings about consequential exclusions such as humans antecedent some biological evolutionary phases (salvation only for the homo sapiens sapiens?), animals, and other forms of life. The problem is the consequential consideration of the excluded as inferior or inferior because they are excluded. The article is about the social, cultural acceptance of the inconsistent coexistence of salvation and exclusion in social cultures and usual ways of thinking. This is not intended here as a theological issue, moreover explicitly treated by some religions. But, rather, as an issue that if philosophically unsolved, not debated, accepted for granted, is compatible with and induces the homo-centrism illusion, egoism, no respect for Nature and supremacist attitudes. Such issue should be interdisciplinary elaborated, theologically induced and oriented with which it can have various levels of compatibility” (pages 2,3).
“The article is about the need to philosophically and interdisciplinary elaborate the inadmissibility of exclusion. The point is the admissibility of the exclusion and how to assume philosophical, cultural, interdisciplinary approaches intrinsically incompatible. Such ethical inadmissibility should be conceived philosophically and interdisciplinary” (page 4).
_____________________
I removed the citations to Peter Abelard and Søren Kierkegaard because, if maintained and elaborated, they would have required to widen the discourse in a misleading way (page 3).

Reviewer 2 Report
Maybe you could be more concrete by giving more examples thus explaining what do you mean by using contemporary understanding of concept.
Author Response
Dear Reviewer,
Thank you for the comments.
I found the expression “contemporary understanding” only in the abstract. I added text.
However, the point is elaborated at page 11 where it’s written:
“New approaches should be considered such as new relativistic understanding of space and time, general networked systemic interdependence, phenomena of acquisitions of new properties such as in phase transitions and emergence, a quantum understanding of the entangled world. Consider, for instance, Quantum Retrocausation [19,20] and other forms of so-called reverse causality, and other theories of the universe.”

Round 3
Reviewer 1 Report
I now understand better the point of view of the author and I think that the paper is ready for publication.